# The Neutrophil-to-Lymphocyte Ratio is Related to Disease Activity in Relapsing Remitting Multiple Sclerosis

**DOI:** 10.3390/cells8101114

**Published:** 2019-09-20

**Authors:** Emanuele D’Amico, Aurora Zanghì, Alessandra Romano, Mariangela Sciandra, Giuseppe Alberto Maria Palumbo, Francesco Patti

**Affiliations:** 1Department “G.F. Ingrassia”, MS Center, University of Catania, Via Santa Sofia 78, 95123 Catania, Italy; aurora.zanghi@studium.unict.it (A.Z.); patti@unict.it (F.P.); 2Department of Surgery and Medical Specialties, Division of Hematology-A.O.U. Policlinico-OVE, Catania, Via Santa Sofia 78, 95123 Catania, Italy; sandrina.romano@gmail.com; 3Department of Economics, Business and Statistics, University of Palermo, 90128 Palermo, Italy; mariangela.sciandra@unipa.it; 4Department “G.F. Ingrassia”, Division of Hematology-A.O.U. Policlinico-OVE, Catania, University of Catania, Via Santa Sofia 78, 95123 Catania, Italy

**Keywords:** neutrophils, lymphocytes, NLR, multiple sclerosis, disease activity

## Abstract

Background: The role of the neutrophil-to-lymphocyte ratio (NLR) of peripheral blood has been investigated in relation to several autoimmune diseases. Limited studies have addressed the significance of the NLR in terms of being a marker of disease activity in multiple sclerosis (MS). Methods: This is a retrospective study in relapsing–remitting MS patients (RRMS) admitted to the tertiary MS center of Catania, Italy during the period of 1 January to 31 December 2018. The aim of the present study was to investigate the significance of the NLR in reflecting the disease activity in a cohort of early diagnosed RRMS patients. Results: Among a total sample of 132 patients diagnosed with RRMS, 84 were enrolled in the present study. In the association analysis, a relation between the NLR value and disease activity at onset was found (V-Cramer 0.271, *p* = 0.013). In the logistic regression model, the variable NLR (*p* = 0.03 ExpB 3.5, CI 95% 1.089–11.4) was related to disease activity at onset. Conclusion: An elevated NLR is associated with disease activity at onset in RRMS patients. More large-scale studies with a longer follow-up are needed.

## 1. Introduction

Multiple sclerosis (MS) is a chronic autoimmune disease of the central nervous system (CNS), in which inflammation, demyelination, and axonal loss coexist, manifesting in a plethora of clinical signs and symptoms [1]. Regarding its pathogenesis, it has been suggested that autoreactive myelin-specific T cells may trigger and modulate the access and trafficking of inflammatory leukocytes to the CNS. Increasing evidence also suggests a fundamental role of B cells in the pathogenesis and development of MS [2].

Neutrophils are bone-marrow-derived cells that represent the most abundant peripheral blood leucocyte and are able to generate extracellular traps, which can be proinflammatory and provide a potential source of autoantigens, triggering autoimmunity [3]. The contribution of neutrophils to CNS autoimmune diseases was first suggested by studies on experimental autoimmune encephalomyelitis (EAE), the animal model of MS, in which neutrophils delayed the onset and decreased the severity of EAE [3,4,5,6].

The role of the neutrophil-to-lymphocyte ratio (NLR) of the peripheral blood in several autoimmune diseases has recently been suggested as a potential cheap and effective surrogate marker for the systemic inflammatory state and thus disease activity [5]. Few studies have addressed the significance of the NLR in terms of being a marker of disease activity in MS [7,8,9].

The aim of the present study was to investigate the significance of the NLR in reflecting the level of disease activity in a cohort of early diagnosed relapsing–remitting MS (RRMS) patients.

## 2. Materials and Methods

The iMED^©^ software (6.5.6, Merck Serono SA, Geneva, Switzerland) was used as the data entry portal, and we followed a rigorous quality assurance procedure to ensure data quality.

The inclusion criteria were: 1) in the age range of 18 to 55 years old; 2) a diagnosis of RRMS according to the 2017 McDonald criteria (retrospectively evaluated at the end of the diagnostic-iter) [10].

The exclusion criteria were: 1) short-term steroid use in the last 30 days; 2) recent infections (≤1 month); 3) stressful concomitant events in the last 6 months (e.g., traumatic bone fractures); 4) a history of tumors; 5) pregnancy; 6) confirmed autoimmune comorbidities (rheumatoid arthritis, psoriasis, Sjögren syndrome, etc.) (Figure 1).

### 2.1. Clinical Data

Data were collected from each patient at the time of enrollment, defined as the first access to a diagnostic MS center prior to setting a confirmed MS diagnosis or starting any disease-modifying therapy (DMT).

The data collected were the following: demographic data, including patient age and gender, clinical data including the date of MS onset, type of MS, number of relapses in the year prior to diagnosis, and level of disability assessed by the Expanded Disability Status Scale (EDSS) in the year prior to diagnosis [11]. As a radiological measure of disease activity, we considered the number of brain and spinal lesions detected on T1 gadolinium (Gd) magnetic resonance imaging (MRI) sequences (Ingenia 1.5 Tesla-Philips^®^). Lesions and imaging were compared with the support of available software (Carestream^®^ Healthcare Information Systems software).

Patients consecutively included in the present study were divided into two groups—low (L) and high disease activity (H), according to the level of disease activity in the year prior to diagnosis, and considering the definition of highly active MS in patients naïve to DMT [12,13,14,15].

High disease activity at onset was classified as: ≥2 relapses in the year prior to study entry and ≥1 Gd-enhancing lesion at the time of study [12,13,14,15].

### 2.2. Blood Tests

Blood cell counts of neutrophils and lymphocytes were obtained via routine blood sampling, as part of the clinical practice in each enrolled patient. The last blood test values from immediately prior to commencement of the first DMT were used, and blood tests taken within 30 days of the start of short-term steroid treatment for relapse were excluded.

Blood sampling was performed between 8.00 and 8.30 a.m. and collected in dipotassium-ethylenediaminetetraacetic acid tubes. Samples were analyzed within 40 min to 1 h of collection. A complete blood count (CBC) was obtained using a blood count machine (ABX micros 60, Horiba medical^®^).

### 2.3. Outcomes

The primary aim of the present study was to investigate, in a real-world setting, whether the NLR is related to high disease activity at the time of disease onset.

### 2.4. Ethical Standards

The study protocol was approved by the Local Ethics Committee (Comitato Etico Catania 1, n.17/2019/PO). All patients provided written informed consent. The present study was conducted in accordance with the ethical principles of the Declaration of Helsinki and appropriate national regulations.

### 2.5. Statistical Analysis

All patient characteristics summary statistics are reported in terms of frequencies (%) for categorical variables, mean ± standard deviation (SD), or median with interquartile range (IQR) for continuous variables. The NLR was calculated as the quotient between the neutrophil and lymphocyte cell counts in blood. The distribution of the NLR was, as expected for a quotient, highly skewed. Therefore, in order to correct for skewness, a log transformation of the NLR was considered. The resulting transformed data show an approximatively Gaussian distribution. Normality assumption has been assessed by using both visual methods (Q-Q plot) and the Kolmogorov–Smirnov (K–S) normality test. Associations between qualitative variables were analyzed using Pearson’s X2 tests. In order to express results as a percentage of the maximum possible variation, the V-Cramer index was used instead of the X2 test. This index can vary between 0 (in the case of independence) and 1 (in the case of maximum association). Several associations were evaluated: between the log-NLR (ln-NLR) and gender, age, EDSS value, and disease activity, respectively. Each variable was treated as categorical according to literature cut-off and the quantile of the observed distributions. In particular, the ln-NLR was dichotomized as 0 for ln-NLR values of ≤0.6 and 1 when ln-NLR was >0.6; gender was defined as 0 for Male and 1 for Female; 40 was the cut-off used for age (0 when age was ≤40 and 1 when age was >40); the EDSS categories were 0 (when the EDSS value was ≤3.5) and 1 (when the EDSS value was >3.5); disease activity as 0 (group L/low disease activity) and 1 (group H/high disease activity). A logistic regression model was used to study the relationship between disease activity (0 = group L, 1 = group H) and the following variables: gender, age (expressed as a continuous variable), ln-NLR expressed as a continuous variable, and the EDSS value. Logistic regression assumptions were verified; the outcome was a binary variable and no high correlations among predictors were observed. All analysis was performed using the SPSS version 21 statistical software (IBM SPSS Statistics 21, IBM, Armonk, NY, USA).

## 3. Results

Among a total sample of 132 RRMS patients, 84 fulfilled the required inclusion criteria and were enrolled in the present study (Figure 1). Of these, 38 were in group L and 46 in group H. Patient demographic and clinical characteristics are shown in Table 1. Patients in group H showed a higher EDSS score at disease onset (*p* = 0.020). Blood tests showed that the ln-NLR ratio was highest in patients in group H (1 ± 0.4 vs. 0.7 ± 0.4, *p* = 0.032) (Figure 2).

The association analysis revealed a V-Cramer of 0.271 with a *p*-value of 0.013 between disease activity and the ln-NLR value (Figure 3). No associations were found between the ln-NLR value and other demographical and clinical variables (Table 2). In the regression logistic regression model, the relation between the highest ln-NLR and disease activitywas maintained (*p* = 0.03 expB 3.5, CI 95% 1.089–11.4) (Table 3).

## 4. Discussion

We found that a higher NLR value increased the risk of disease activity in our cohort. The NLR is employed as a marker of systemic inflammation in other autoimmune diseases and cancers, since it is a low-cost analysis [5,6]. A study using the EAE animal model of MS showed that neutrophil depletion led to a reduction in the severity of the disease [16].

Neutrophils are short-lived, bone-marrow-derived cells, and are the most abundant peripheral blood leucocytes. The number and lifespan of neutrophils are under tight control. Neutrophils are phagocytic and microbiocidal, releasing tissue-remodeling enzymes and reactive oxidative species, as well as proinflammatory cytokines and chemokines, which can provide a potential source of autoantigens, triggering autoimmunity [3,5]. In other fields, such as rheumatoid arthritis, several studies have suggested that joint destruction and disease activity are directly correlated with the recruitment of neutrophils in the synovium [17].

Few studies have investigated the association of the NLR with disease activity in MS patients in a real-world setting. The role of neutrophil counts in MS is a recent and interesting challenge, focusing on the involvement of the innate immune system [18].

Findings are available for MS and optic neuritis (ON) patients as compared with healthy controls (HC) in a large Danish cohort. The NLR was found to be higher in MS and ON patients as compared to HC, indicating the occurrence of chronic inflammation. The NLR may be an inexpensive and easily accessible supplemental marker of disease activity in RRMS patients, revealing an association between a high NLR value and MS occurrence as compared with controls [7,9,18]. Furthermore, a higher NLR during relapse as compared to that during remission was shown in a study by Demirci et al., in which the NLR predicted activity with 67% sensitivity and 97% specificity [7], indicating the value of NLR use in measuring disease activity [7]. Furthermore, NLR has also been investigated as a possible marker relating to the depression, anxiety, and stress score in MS patients [19].

Naegele et al. demonstrated that the increase in neutrophil count in RRMS patients is most likely due to a decrease in apoptosis, and that neutrophils have an altered cell surface expression of certain proteins, which may enhance recruitment to sites of inflammation [4].

However, there are a lack of available data regarding the prediction of long-term disease progression, according to a high NLR ratio. Two previous studies found that the NLR correlates with the EDSS value [7,8], whilst another found no association [18]. Interestingly, Hemond et al. investigated the association of the NLR ratio with clinical, neuroimaging, and psycho-neuro-immunological associations in a large cohort of MS patients, also including patient-reported outcomes. Here, NLR strongly predicted increased MS-related disability independent of all demographic, clinical, treatment-related, and psychosocial variables (*p* < 0.001) [20]. Moreover, Giovannoni et al. failed to find a correlation between the mean levels of several proinflammatory immunological markers and clinical disease progression [21]. Therefore, to further clarify whether the NLR is associated with MS disability, studies with long follow-up times are needed, since this would enable measurement of disease progression. The use of a marker obtained from a simple routine blood exam could make the NLR ratio a useful supplemental biomarker of disease activity in MS patients in clinical practice.

The present study has some limitations, many of which are related to the retrospective design and small sample size. The results build on simple data from routine laboratory tests, and, therefore, may be biased toward the availability of certain laboratory data. Furthermore, there exists no data regarding concomitant diseases that could alter the neutrophil count, or other data such as smoking status. Prior to any possible use of the NLR in clinical practice, more data, preferably from multi-center, long-term studies, are needed. Studies comparing the NLR with MRI and cerebrospinal fluid markers would also be desirable. Additionally, the impact of disease-modifying drugs on the NLR is required to establish the clinical application of this parameter. The NLR has gained academic relevance, contributing to the knowledge of MS immunology.

## Figures and Tables

**Figure 1 cells-08-01114-f001:**
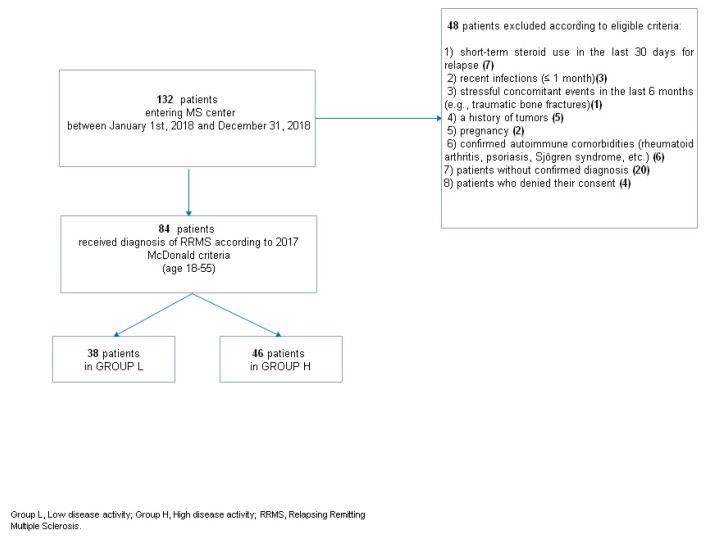
Patients’ selection flow chart. RRMS; relapsing–remitting multiple sclerosis.

**Figure 2 cells-08-01114-f002:**
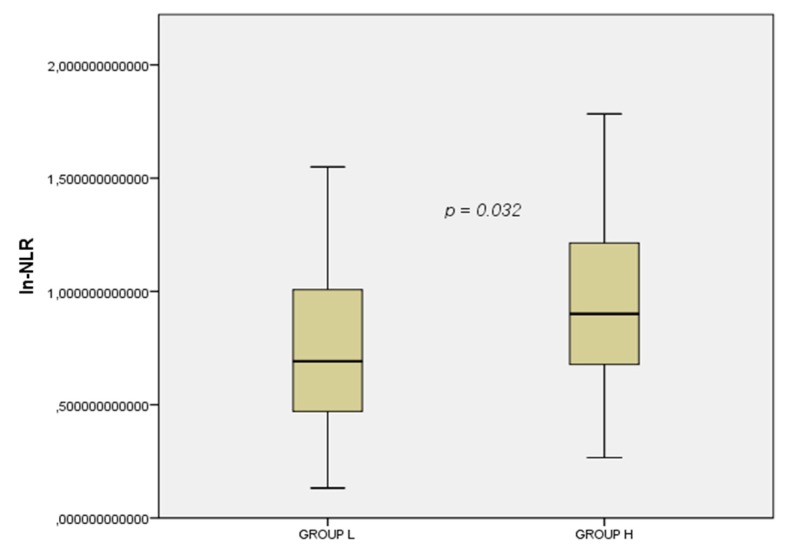
Box plot of the ln-NLR in the two groups. Group L, Low disease activity; Group H, High disease activity; ln.NLR, log transformation of the neutrophil to lymphocyte ratio.

**Figure 3 cells-08-01114-f003:**
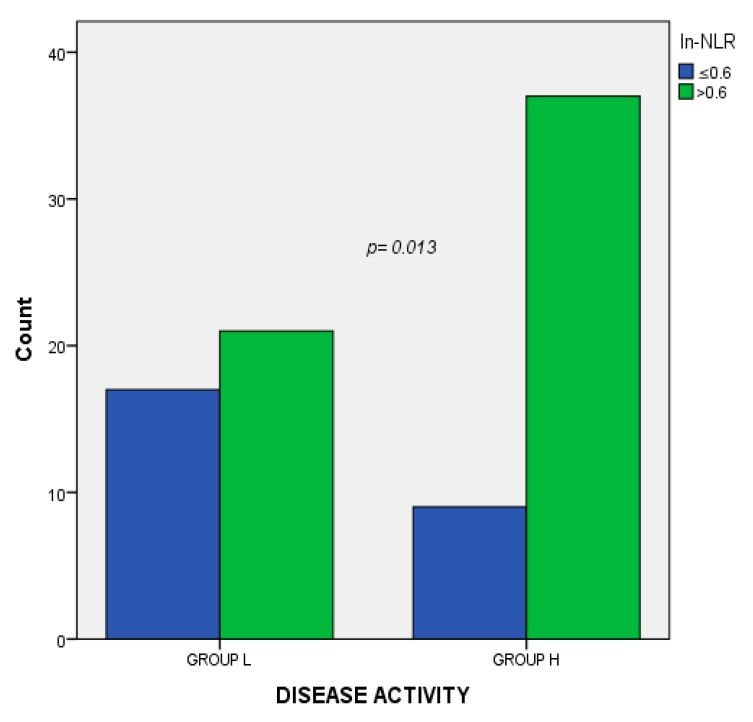
Association analysis between disease activity and the ln-NLR. Group L, Low disease activity; Group H, High disease activity; ln.NLR log transformation of the neutrophil to lymphocyte ratio.

**Table 1 cells-08-01114-t001:** Demographical and clinical characteristics of the two groups at disease onset.

	Group L (*n* = 38)	Group H (*n* = 46)	*P*-Value
Men	19 (50%)	19 (41.3%)	0.42
Women	19 (50%)	27 (58.7%)	
AGE	43 ± 13.4	36.9 ± 12.5	0.23
L%	29 ± 8.5	27.8 ± 6.6	0.95
N%	62.3 ± 7.7	63.3 ± 7.7	0.57
lnNLR	0.7 ± 0.4	1 ± 0.4	*0.032*
EDSS value	1.5 (1.0–3.0)	2.5 (1.0–5.0)	*0.020*

Results are expressed as mean ± SD, median (IQR) and No. (%). IQR = interquartile range; SD = standard deviation. Abbreviations: EDSS, Expanded Disability Status Scale; L, lymphocytes, N, neutrophils; ln-NLR, logarithmic-neutrophils/lymphocyte ratio.

**Table 2 cells-08-01114-t002:** Association analysis between the ln-NLR and demographical/clinical parameters.

Patient Characteristics vs. ln-NLR	V-Cramer Index	*P*-Value
Gender	0.016	0.289
Age	0.052	0.637
Disease activity	0.271	*0.013*
EDSS value at onset	0.062	0.567

**Table 3 cells-08-01114-t003:** Logistic regression model for disease activity at onset.

	Exp B	*P*-Value	Confidence	Interval (95%)
			Lower	Upper
Gender	0.887	0.778	0.352	2.18
Age	1.00	0.555	1.00	1.00
ln-NLR	3.5	0.03	1.08	11.4
EDSS value at onset	1.00	0.926	1.00	1.00

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
