# Peer review of "The Neutrophil-to-Lymphocyte Ratio is Related to Disease Activity in Relapsing Remitting Multiple Sclerosis"

_cells, 2019, doi:10.3390/cells8101114_

Round 1
Reviewer 1 Report
The authors attempt to ascertain whether neutrophil:lymphocyte ratio reflects disease activity in MS. The manuscript is poorly written with significant lack of detail in describing methodology.
Major issues
It is not clear why they divided the subjects into the groups as described or the rationale for this choice. A graph of NLR in these two groups is required. How was disease activity defined and what disease activity (clinical or radiological) are they referring to when describing results of correlation and the regression models? Currently it is not clear to me whether NLR is related to clinical or radiological activity or both and it would be interesting to graphically depict the correlation described (to ensure that this is not driven by outliers).
Minor concerns
This manuscript needs extensive English editing.
Author Response
Please see all the attachment.

Reviewer 2 Report
General comments: It is an interesting manuscript that assesses the relationship between disease activity and neutrophil-to-lymphocyte ratio (NLR) in a cohort of 84 patients with multiple sclerosis (MS). Admitting the few available data, this work may add some value to the current knowledge in this field. NLR seems to predict the disease activity, but did not significantly correlate with physical disability scores. The manuscript has several limitations, some of which were already adequately addressed in the discussion. However, other limitations (mainly related to the adapted method and its brief description) requires to be addressed. The manuscript is comprehensible in the actual form, but would benefit from minor editing. For other comments, please see below:
Introduction:
Reference 7 does not include any information on NLR as mentioned in the citation.
Methods:
Pleasure provide a proper reference for the 2017 revised McDonald criteria (Thomspon et al., Lancet Neurol. 2018) and for the expanded disability state scale (Kurtzke, 1983).
In the inclusion criteria, it is only mentioned that adult patients with MS took part of the study. Did the authors take into consideration possible active infections, active or chronic inflammatory diseases (e.g. inflammatory bowel disease and Sjögren's syndrome), other autoimmune diseases, pregnancy, present malignancy and surgery within the preceding three months, stressful events?
Radiology part: For a research paper, it is important to provide the technical details of the MRI acquisition (name of the machine and the manufacturer, description of the employed technique to detect new lesions, etc.). In addition, “gadolinium lesion on T1” may be substituted with “gadolinium enhanced lesions on T1’.
Hematology part: As seen in the neuroimaging part, the actual paragraph does not allow for future replication of the current results. Please add more details on the technique (e.g., how the venous sample was collected (e.g., in the dipotassium-Ethylenediaminetetraacetic Acid tubes), the time of the day, the time between collection and analyses (e.g., 1 h), the name of the manufacturer,).
It is crucial to mention if the radiology and hematology data were obtained using the same standardized tool (if that is the case) in all patients. Otherwise, the technical difference may render it difficult to interpret the data.
It is important to mention when data were collected. As the one can assume, the authors were interested in the clinical and radiological data prior to setting MS diagnosis, and in the hematological data prior to initiating disease-modifying therapies? EDSS score appear to be collected at ‘disease onset’ as per p.3 l. 93 and at and ‘at diagnosis’. The time-line of data collection need to be clarified. In addition, the criteria that are used to classify patients in group 1 and 2 are very briefly stated and require further justification and elaboration.
Statistical analysis:
It is important to state the data did not follow normal distribution (e.g., tested by means of Shapiro-Wilk or Kolmogorov-Smirnov or other tests) which (probably) justifies the rationale for adapting the nonparametric tests. For the group comparison, a non-parametric comparison of two independent samples is usually done using Mann-Whitney test. Wilcoxon test is usually spared for comparing two dependent samples. Please explain. It is important to state whether the assumptions for running logistic regression analysis were respected. There is no mention of correlation analysis in this part. This should be clarified (the number of variables of interests) along with the used test (i.e., Spearman’s correlation coefficient?), and the considered patients (i.e., the whole cohort?). The reader will discover that it was done when reading the results.
Results:
Please add exact p values for group comparison. It would be also helpful to add a graphical data representation and add a table reporting the results of logistic regression.
The discussion may benefit from adding relevant references:
Hemond CC, Glanz BI, Bakshi R, Chitnis T, Healy BC. The neutrophil-to-lymphocyte and monocyte-to-lymphocyte ratios are independently associated with neurological disability and brain atrophy in multiple sclerosis. BMC Neurol. 2019 Feb 12;19(1):23. Al-Hussain F, Alfallaj MM, Alahmari AN, Almazyad AN, Alsaeed TK, Abdurrahman AA, Murtaza G, Bashir S. Relationship between Neutrophil-to-Lymphocyte Ratio and Stress in Multiple Sclerosis Patients. J Clin Diagn Res. 2017 May;11(5):CC01-CC04.
Round 2
Reviewer 1 Report
This is an improved manuscript - still requires some English editing and spell check.
Please change Thompson criteria to 2017 McDonald Criteria.
Author Response
This is an improved manuscript - still requires some English editing and spell check.
Please change Thompson criteria to 2017 McDonald Criteria.
Thank you reviewer, we have edited language with an expert team of text editing; we have also changed accordingly Thompson criteria into McDonald.
Reviewer 2 Report
The referee would like to thank the authors for the attention they paid to the comments and suggestions. The manuscript improved in its revised versions. Some comments remain.
It is now clarified in the discussion that some data followed normal distribution (presented as mean ± SD) and others did not (presented as median (IQR)). The authors states that Non-normally distributed variables were analyzed using the Mann–Whitney U-test. However, it is not that clear whether the authors adapted a nonparametric test (i.e., Mann-Whitney test) for all the continuous variable, or if they alternately used nonparametric (i.e., Mann-Whitney test) and parametric (e.g., Student t test, Welch t test, ANOVA) tests as appropriate.
The authors did not address these previous comments: It is important to state whether the assumptions for running logistic regression analysis were respected. Also, regarding correlation analysis, they still did not mention the number of variables of interests (i.e., age, EDSS), and whether the correlation analysis was run in the whole cohort or in one of the two groups. The authors do not precise whether they applied or not statistical correction (Family Wise Error or False Discovery Rate approach) for correlation analysis. If they did not, this should be acknowledged among the limitations since it might increase the change for type I error (false positive).
In addition, correlation analysis is usually done for continuous variables. It is therefore important to explain how this was done for categorical variables (disease activity and gender). For categorical variables, it is usually advisable to use two categories (e.g., 1 for men, 2 for women, or 1 for high disease activity, 2 for low disease activity) and perform correlation analysis. An accepted way to analyze the relationship between categorical independent variables (i.e., gender and disease activity in this study) and dependent variable (i.e., ln-NLR) is by performing group analysis comparing the dependent variable between patients with high and low disease activity (as it is already done in the manuscript) and between men and women.
Regarding the correlation analysis results, it is stated that : “In the correlation analysis, a correlation between group H and highest ln-NLR value was found 119 (rho 0.261, p = 0.008)”. The sentence should state the correlation between the disease activity at onset in group H and ln-NLR as the one can read in table 2. The authors may want to omit the correlations with categorical variables based on the previous comment, especially the relationship between the disease activity and ln-NLR is already addressed by a group comparison and a binary logistic regression.
In the discussion, when the authors review the available literature on the correlation between EDSS and NLR, with inconsistent studies, it would be interesting to comment on their current result in the light of these studies. In the actual study, EDSS scores were significantly higher in group H (p=0.02), EDSS scores tended to correlate with NLR (p=0.084), but they did not predict disease activity in the binary regression model (0.926).
Minor editing: In figure 1, please correct « partecipate. » and define Group L and H in the figure legend. In the discussion, please add a space before the reference in the following sentence: “Furthermore NLR has been investigated also as a possible 152 markers related with Depression, Anxiety and Stress score in MS patients(19).” There is no need to redefine the acronym EDSS in l. 158 of the discussion.
